# Characterization of *Bacillus cereus* in Dairy Products in China

**DOI:** 10.3390/toxins12070454

**Published:** 2020-07-14

**Authors:** Xiao-Ye Liu, Qiao Hu, Fei Xu, Shuang-Yang Ding, Kui Zhu

**Affiliations:** 1College of Veterinary Medicine, China Agricultural University, Beijing 100193, China; xiaoyeliu@pku.edu.cn (X.-Y.L.); sy20193050810@cau.edu.cn (Q.H.); 2Department of Mechanics and Engineering Science, College of Engineering, Academy for Advanced Interdisciplinary Studies, and Beijing Advanced Innovation Center for Engineering Science and Emerging Technology, College of Engineering, Peking University, Beijing 100871, China; 3National Center for Veterinary Drug Safety Evaluation, College of Veterinary Medicine, China Agricultural University, Beijing 100193, China; dingsy@cau.edu.cn; 4National Feed Drug Reference Laboratory, Feed Research Institute, Chinese Academy of Agricultural Sciences, Beijing 100081, China; xufei@caas.cn

**Keywords:** *Bacillus cereus*, China, dairy product, prevalence, virulence factor

## Abstract

*Bacillus cereus* is a common and ubiquitous foodborne pathogen with an increasing prevalence rate in dairy products in China. High and unmet demands for such products, particularly milk, raise the risk of *B. cereus* associated contamination. The presence of *B. cereus* and its virulence factors in dairy products may cause food poisoning and other illnesses. Thus, this review first summarizes the epidemiological characteristics and analytical assays of *B. cereus* from dairy products in China, providing insights into the implementation of intervention strategies. In addition, the recent achievements on the cytotoxicity and mechanisms of *B. cereus* are also presented to shed light on the therapeutic options for *B. cereus* associated infections.

## 1. Introduction

*Bacillus cereus* is a Gram-positive, endospore-forming, foodborne pathogenic bacterium that is widely distributed in the natural environments, frequently found in foods especially dairy products and even persisting in host epithelial cells [1,2,3,4,5,6,7]. As an opportunistic pathogen, *B. cereus* has long-term emerged as a health threat to humans and animals, involving both domestic and wild animals [8,9,10,11]. Foodborne outbreaks involving *B. cereus* in China usually occurred through dairy products [12,13,14,15]. Major symptoms of food-poisoning caused by *B. cereus* are divided into either diarrhea or emesis [5,16]. Diarrhea is mainly induced by three enterotoxins that belong to the family of pore-forming toxins (PFTs) [17], including non-hemolytic enterotoxin (Nhe) [18,19], hemolysin BL (Hbl) [20] and cytolysin K (CytK) [21], while the emetic syndrome is tightly connected to a lethal toxin known as “cereulide”, which is synthesized by a non-ribosomal peptide synthetase (NRPS) encoded by a *ces* gene [22,23].

Most toxins of *B. cereus* belong to the family of pore forming toxins (PFTs); among them, Nhe and Hbl are similar to the well-known cytolysin A (ClyA) of α-PFTs family, while CytK and hemolysins are members of β-PFTs [17,24,25]. PFTs have the capacities of altering the plasma membrane permeability of target cells, potentially leading to cell death and triggering the signaling pathways [26,27,28]. Nhe and Hbl have a similar mechanism of action; the three components of Nhe complex display the highest cytotoxicity at a ratio close to 10:10:1 for NheA, B and C [29,30]. In terms of Hbl, the ratio at L2: L1: B = 1:1:10/ 10:1:10 prompts the most rapid pore formation [20]. Recent studies also found that LITAF and CDIP1 work as the Hbl receptor [31]. The signaling pathways triggered by Nhe, as shown in Figure 1a, include the induced cell apoptosis though ASK1 and Fas-p38 MAPK mediated caspase-8 dependent pathways [32]. In addition, the most recent research suggested that Nhe and Hbl operate synergistically to activate the NLRP3 inflammasome and induce inflammation [33]. Moreover, Nhe have the concerted action with sphingomyelinase in pathogenic *B. cereus* to cause full virulence and formation of disease [34]. The two caspase-1 dependent inflammatory pathways triggered by Nhe include the form of inflammation initiated by IL-1β release and pyroptosis induced by the activation of GSDMD (Figure 1a). Moreover, the emetic toxin, cereulide is a K^+^ ionophore toxin that damages the cellular membrane potential through inhibiting the synthesis of RNA [35], affecting mitochondrial function, resulting in expansion of mitochondria and formation of vacuoles in the protoplasm of sensitive cells, inducing cell apoptosis and even fulminant liver failure [36,37] (Figure 1b). Moreover, cereulide is terribly unwholesome and could accumulate in multiple organs [38]. CytK and hemolysins otherwise do harm to the target membrane, bringing cell lysis and apoptosis in macrophages [39] (Figure 1c).

Besides, *B. cereus* can survive in the gastrointestinal tract with versatile virulence factors [40,41]. Therefore, many other infections associated with *B. cereus* have been reported including meningitis, brain abscess [42], cellulitis [43], endophthalmitis [44,45], pneumonia [46], endocarditis [47] and osteomyelitis [48]. More seriously, persistent *B. cereus* strains, which are highly detrimental pathogenic bacteria against antibiotic therapies, were also found in patients in the USA [49]. Altogether, tracing the source of *B. cereus,* such as in dairy products, is of particular concern.

At present, to the best of our knowledge, there is no detailed information focusing on the potential risk of *B. cereus* from dairy products in China, not to mention the summaries about the detection and toxicity mechanisms of *Bacillus* virulence factors. In the current review, we summarized the demand for dairy products in China, the prevalence and detection of *B. cereus* and the virulence factors. We aim to give an overview of *B. cereus* in dairy products, which may contribute to the implementation of effective strategies to prevent and control foodborne pathogenic *B. cereus* in dairy products.

## 2. Dairy Products with *Bacillus cereus*

Dairy products are excellent nutrition for both young animals and human beings [50,51,52]. Notably, milk contains all eight kinds of essential amino acids, minerals, vitamins, and fatty acids with optimal proportions of nutrients [8,9]. To satisfy the increasing needs of human, milk associated dairy products are derived from diverse sources, which are known to fall into various categories including liquid milk, milk powder, cheese, condensed milk, milk fat and ice cream [51,52,53]. Demand for dairy products varies sharply from one region to another [54]. For instance, liquid milk had a top priority for Chinese customers in contrast to American and those from other countries, the proportions being 55% in China and 25% in America, respectively, while cheese had the smallest demand (just 13%, yellow part) in China compared with other countries (Figure 2a). In addition, dairy products in the Chinese market have been in short supply, and the export volume of these was far less than the import amount in China from the year of 2013 to 2018 (Figure 2b). According to the phased consumption targets of the National Food and Nutrition Advisory Committee, per capita consumption of milk will reach 28kg by 2020 and 41kg by 2030 [53]. Likewise, the sales of milk in China will increase from 119.5 to 128.3 billion yuan from 2018 to 2022, with an average annual compound growth rate of 1.7%, and the size of the milk market will sustain a steady growth, as put forward by the China Business Research Institute. Consequently, the sale volume of milk in China will gradually augment, and the market prospect is considerable [55,56]. In view of the high demand for dairy products and the outbreaks of bacterial contamination in liquid milk and milk powder in China, it is urgent to carry out effective analytical tests, especially the detection of microbes during manufacturing, selling and importing of milk [12,57,58,59]. In addition, the uncertain microbial growth rate and toxicity of microbial metabolites all threaten food safety [60,61,62,63]. As shown in Figure 2c, although preventive measures have been made to manage the contamination of dairy products in the last century [64], the outbreaks of *B. cereus* spp. contamination constantly arose in various milk products all over the world [61,62,65,66,67,68]. The prevalence of *B. cereus* in dairy products is difficult to estimate, and food poisoning incidents caused by *B. cereus* still remain a thorny problem worldwide due to the high tolerance of *B. cereus* to various environments and strong propagation capacity of *B. cereus* spores [2,5,13,47,69]. The *B. cereus* isolates correspondingly are suspected of threatening the safety of raw milk and dairy products in China [5,13,70]. Therefore, we will next focus on the pattern of *B. cereus* contamination in dairy products in China.

## 3. Prevalence of *Bacillus cereus* from Dairy Products in China

### 3.1. Contamination of *Bacillus cereus* Isolates

*B. cereus* prevails in soil and dairy farms and often pollutes foods like raw milk and all dairy products [3,71]. Spores of *B. cereus* can primarily spread through soil and air [69]. Researches showed that 1g of soil contains 50–380,000 CFUs (colony-forming units) *B. cereus* spores, and 1 m^3^ of air has at least 100 CFUs of *B. cereus* spores. Thus, the abundant *B. cereus* group spores in the environment are a major cause of the high prevalence rate of *B. cereus* [5,72]. *B. cereus* and its spores subsequently have a great opportunity to circularly contaminate dairy farms, human market, food supply places, and dairy products and colonize the intestinal tract of invertebrates and cause illness in humans afterwards (Figure 3a). The Centers for Disease Control (CDC) website claimed that there were 619 confirmed outbreaks of *Bacillus*-related poisoning from 1998 to 2015 [2,73]. Specifically, the diarrheal illness caused by *B. cereus* is often related to meats, milk, vegetables and fish, while the emetic type is most possibly associated with rice products [15]. Previous studies have shown that *B. cereus* isolated from raw milk have the ability to remain active after pasteurization or ultra-high temperature (UHT) sterilization, which ensues in bacterial pollution in the final products [12,66,73,74]. Thus, the health hazards originated from *B. cereus* in milk industry in China require rapid and proper handling.

### 3.2. Distribution of *Bacillus cereus* in Milk and Milk Products in China

The prevalence of *B. cereus* in China has distinct traits, owing to the specific market demands and physical differences. Liquid milk and milk powder are consumed by the largest part of the population in China (Figure 2a). It is quite a coincidence that a high prevalence rate of *B. cereus* appears in liquid milk (44%) and milk powder (26.1%) (Figure 3b). A recent report claimed that *B. cereus* were widely present in pasteurized milk in China, showing that 100 of *B. cereus* isolates are distributed in most Chinese cities including Hong Kong, Guangzhou, Shenzhen, Harbin, Ningxia, Beihai, Hai kou, etc. [14]. In general, there is a relatively lower prevalence rate of *B. cereus* in dairy products in southern China than in the northern region with exceptions (Table 1 and Figure 3c**)**. For instance, infant formula in the Liaoning province (42%) was contaminated with *B. cereus* more seriously than that in Yunnan province (12%) (Table 1). The regular rates of *B. cereus* in other cities or provinces, such as Beijing, Liaoning, Gansu, Yunnan, northeast China, were found to be 30%, 27%, 19%, 10% and 16%, respectively [13,75,76,77]. Among these data, Yunnan province and the Northeast China indeed had a relatively low prevalence of *B. cereus* in powdered infant formula (PIF). Dairy products in Gansu province was polluted by *B. cereus* to a moderate extent (19%). The most striking data in Table 1 were observed in Beijing and Liaoning province, and the investigation conducted in major cities of China during 2011–2016 suggested that approximately 27% of the pasteurized milk contained *B. cereus,* and the contamination of *B. cereus* was 31% (11/36) in northern China and 25% (33/132) in southern China. Concerning infant formula in the Chinese market, two reports were produced in 2012–2013 and 2015, according to which 14% and 42% of formula contained *B. cereus*, respectively. It was also revealed that 8.2% of PIF samples in China were contaminated with *B. cereus* strains [78]. Overall, the regional characteristics of the prevalence of *B. cereus* in dairy products in China cannot be clearly defined by latitudes or longitudes. More scientific research into the epidemiological nature of *B. cereus* in milk is worth pursuing. From an international perspective, dairy products from African nations are more likely to be polluted by *B. cereus*, which is aided by the poor sanitary conditions [74]. A nationwide survey conducted in America manifested that a total of 18 (8.9%) of 202 samples were positive for presumptive *B. cereus* using the MPN technique (<10 to 50 CFU/mL), which cannot be directly compared with those of most other studies [79]. Remarkably, ice cream tested in Bavaria, Germany, was found to carry *B. cereus* with a high rate of 62.7%, and artisan cheese sold in Mexico had a rather low rate of 28.4% [80]. Nevertheless, no certain epidemiological profile can be obtained from the scanty information on dairy products, domestically and globally. Therefore, it is urgent for countries to address the prevalence of *Bacillus cereus* in their territory.

## 4. Virulence Factors of *Bacillus cereus* and Detection Techniques

Many diseases caused by *Bacillus* groups, such as bovine mastitis, are associated with the virulence factors [40,89,90]. Therefore, it is necessary to further detect the virulence factors secreted from *B. cereus* isolates in a clinical setting. In particular, the analytical assays of virulence factors in a clinical setting would be enormously beneficial to understand the pathogenic mechanism of virulence factors of *B. cereus* and to develop effective treatment strategies.

### 4.1. Virulence Factors of *Bacillus cereus*

The PlcR regulator in *B. cereus* is a transcriptional regulator that controls some of the most known virulence factors. It activates gene expression by binding to a nucleotidic sequence called the “PlcR box” [16,90,91]. PlcR regulator is mainly responsible for the transcription of the genes of metalloproteases (InhA2 and Enhancin), hemolysins (CLO and CytK), enterotoxins (Haemolysin BL, Hbl and Nonhemolytic enterotoxin, Nhe) and phospholipases (PI-PLC, PC-PLC and SM-PLC) [24,92]. Another emetic toxin of *B. cereus*, cereulide, whose synthesis is independent of PlcR, belongs to the Spo0A-AbrB regulon [93]. Cereulide is encoded by the 24-kb cereulide synthetase gene (*ces*) cluster that located on a megaplasmid of pXO1 [94,95].

### 4.2. Detection of *Bacillus cereus* Isolates 

*B. cereus* is a ubiquitous Gram-positive, aerobic or facultative anaerobic, endospore-forming, rod-shaped bacterium [13,71]. The detection and isolation of *B. cereus* strains are mainly based on the colony count technique ISO 7932 [96]. *B. cereus* or presumptive colonies of *Bacillus* are counted on the varieties of *Bacillus* agar by spiral-plating or spread-plating techniques, most probable number (MPN) method and so on [97]. Both *B. cereus* cells and spores in the examined products can be counted according to the colony-forming units (CFUs) [5,98,99]. In addition, *B. cereus* isolates have the capacity for casein, starch and tributyrin hydrolysis as well as lactose fermentation, which inspired the invention of chromogenic medium for *B. cereus* [100,101]. Instead, the identification or analysis of *B. cereus* isolates also can use the PCR, the quantitative real-time PCR by targeting the *16S rRNA* gene [102,103], *groEL*/*gyrB* genes [104] and *panC* gene [105,106,107], or cross-priming amplification [108] and so on.

### 4.3. Detection of Toxins Secreted from *Bacillus cereus*

PCR, RT-PCR and multiplex PCR are the major analytical techniques that are reported in research articles to identify the virulence factors by detecting toxin genes in *B. cereus* (Figure 4a). As we know, the tripartite enterotoxins–Nhe complex consists of NheA, NheB and NheC that were encoded by *nheA*, *nheB* and *nheC* genes [19], as well as the components of Hbl-L2, L1, B were encoded by *hblA*, *hblC* and *hblD* genes separately [21]. Thus, Nhe and Hbl were usually recognized though PCR by targeting their toxic genes [109]. Clinically, a higher rate of *nheA, B, C* genes than that of *hblA, hblC and hblD* genes was disclosed in *Bacillus* samples (Table 1). Other toxin genes such as *cytK* and *cytK-2* genes of CytK, *hly* gene of hemolysins or *cesB* gene of cereulide are also served as the main approaches to determine the positive strains of *B. cereus* by PCR [23,110,111,112,113,114,115,116]. Furthermore, another detection tool, Enzyme Immunoassay (EIA), is employed for the direct inspection on the protein level of the toxic components of *B. cereus* by targeting specific mAbs [117,118], which are able to purify toxins such as NheB and the NheB-C complex and neutralize the cytotoxicity [21,119].

Liquid chromatography–mass spectrometry (LC-MS) and matrix assisted laser desorption/ionization-time of flight (MALDI-TOF) analysis are often used for the rapid detection of emetic *Bacillus* isolates in food products, by analyzing ribosomal subunit proteins [120] or targeting the distinct molecular of cereulide [70,121,122,123,124]. Cereulide is a cyclic molecule composed of a 36-membered ring and a small heat and acid stable cyclic dodecadepsipeptide of 1165 Da, which consist of alternating ester and amide bonds and the structure [-_D_-O-Leu-_D_-Ala-_L_-O-Val-_L_-Val-]_3_ [23]. Besides, cereulide belongs to the surfactin-like peptides and is biosynthesized via nonribosomal peptide synthesis (NRPS) [94]. Recent reports revealed that the identification of cereulide from bacterial extracts peak at *m*/*z* 1191 with a limit of detection (LOD) of 30 ng/mL [121].

Normally, cytotoxicity tests or cell culture methods are used for evaluating the virulence of *B. cereus* isolates. The cytotoxicity of the complexes of diarrheal *Bacillus* enterotoxins accounts for over 90% of the total toxicity [109]. A study showed that Vero and primary endothelial cells (HUVEC) were most sensitive to Nhe, whereas Hep-G2, Vero and A549 cell lines were highly susceptible to Nhe and Hbl. CytK exhibited the highest toxicity on CaCo-2 cells [40], and the emetic toxin cereulide prevented cell proliferation in HepG2 cells by 2 nM [35], also causing vacuolation in HEp-2 cells [125,126]. In general, *B. cereus* toxins exhibited a wide cytotoxicity to those epithelial cells [127] and some toxins also acted on immune cells [39,128].

### 4.4. Analysis of Bacillus Toxin Detections in Dairy Products in China

*B. cereus* and its virulence factors are frequently present in dairy products in China [12,13,14,15,57,58]. This not only impacts the quality of dairy products but also potentially impairs human health. Since there is a huge demand for dairy products among Chinese people of all ages [50,53], once *Bacillus* contamination has occurred, the scope of the damage is extensive and inestimable.

Currently, there is a large number of potential toxin genes related to diarrhea in these *B. cereus* strains, including the genes of *hcbl*, *nhe*, c*ytK* and enterotoxin FM (*entFM*), as well as potential enterotoxins *hlyII* and enterotoxin T (*BceT*) [14]. As shown in Table 2, a study conducted in 10 local dairy farms in Beijing suggested that the *nhe*, *hbl*, and *ces* genes were detected at the rate of 100%, 79.3%, and 1.1%, respectively [13]. Meanwhile, a research involving 12 provinces in China showed that the average virulence gene number in powdered milk was 5.71, and no comparison of the distribution of those genes between different provinces was made [129]. It is logical to conclude that the virulence factors of *B. cereus* can exert influence on the quality of dairy products during their processing, transporting and selling. Still, like *B. cereus*, no striking feature of the distribution of their virulence factors in diverse provinces could be obtained. Similarly, high numbers of isolates carried *nheA* (84.1%), *nheB* (89.9%), *nheC* (84.1%), *hblA* (59.4%), *hblC* (44.9%), *hblD* (53.6%) and *cytK* (53.6%) genes in the production chain of milk in Brazil [62], which is notably bigger than the rate found in Ghana [73]. It is really challenging to sum up the rule of distribution of *Bacillus* toxins in dairy products in China, and this is limited by the deficient detection methods and shortage of useful data, as the notion of *Bacillus* contamination in dairy products only came into view in recent years.

To produce a more forthright analysis of the *Bacillus* toxins, we collected the prevalence of the toxin genes in *Bacillus* samples from pasteurized milk and milk powder as they are the most popular dairy products in China [12,13,14,15,53,57,82,112] (Figure 3a). The data showed that the genes of *nhe A B C* were found in almost all the *Bacillus* isolates from pasteurized milk, while at least 80% of *nhe A* and approximate 100% *nhe B* and *C* in milk powder (Table 1 and Table 2 and Figure 4b). More genes of *hbl A C D* were discovered in pasteurized milk (45%) than in milk powder (36.8%). Similarly, the positive rate of *cytK* genes in pasteurized milk and milk powder was 73% and 44.7%, respectively. The *cesB* gene was mainly found in milk powder (2.6%), and *hlyII* gene only grew in pasteurized milk (54%) (Figure 4b).

However, the detection techniques of *B. cereus* and its virulence factors are strongly limited [5,110,114], and *B. cereus* diagnostic method is still a field to be developed [135]. The widely used nucleic acid-based detection technology cannot accurately determine the bacterial activity and toxin expression [110,114]. Thus, the data analysis was incomplete since it is only based on the occurrence of toxin genes, while the level of protein expression and its toxic effects stay implicit. Therefore, the diversity and perfection of detection techniques of *B. cereus* and its virulence factors are necessary to improve the safety of dairy products in the future. Beyond that, the toxicity mechanisms of *Bacillus* virulence factors deserve to be illustrated, as host cells have frequently interacted with *B. cereus* infection [1,31,33,136,137]. Thus, comprehending the cytotoxicity of *B. cereus* is greatly beneficial in providing the therapeutic strategy for related illnesses caused by intaking *Bacillus*-contaminated dairy products. We believe that the summary of detection techniques of *B. cereus* will certainly be helpful for a more accurate examination and evaluation in the future.

## 5. Conclusions and Future Perspectives

The awareness of food safety in China has risen significantly with the mounting needs for high-quality foods. However, there still exist food poisoning incidents caused by bacteria like *B. cereus* in dairy products [5,12,13,70]. *B. cereus* group is an opportunistic spore-producing pathogen that causes food poisoning with symptoms of vomiting and diarrhea, exhalation of toxins that are the main culprit of damaging liver tissue and inflammatory diseases such as gastroenteritis and meningitis [16,17]. The infectious bacteria were the main focus of some widespread epidemics in history, and therefore, the safety of dairy products should not ever be ignored. In the past decades, although we have made arduous efforts to ensure food safety, the contamination of dairy products with *B. cereus* is still an issue in China. With regard to the average level in China, the investigation carried out from 2011 to 2016 unveiled that about 27.1% of the pasteurized milk on shelf were infested with *B. cereus* and also that the environments of milk production, handling and processing could introduce *B. cereus* into milk products. Together, these assessments implied the high prevalence of *B. cereus* and existence of potential hazards in contaminated pasteurized milk (Figure 3 and Table 1).

On the other hand, *Bacillus* strains can also be used as human probiotics [138], and this field is gaining greater attention [90,139], as *B. cereus* strains or spores serve as probiotics for human use [140,141]. Some countries even utilize their connection with the dairy chain as a source to culture novel probiotic products [142]; however, the consequences could be unfavorable. In contrast to chemical drugs that quantitatively decrease or remain unchanged after metabolic process, the variation trend of the amounts of microbes like *B. cereus* in probiotics is undefined and could even rise exponentially.

In this review, we summarized the risk of *B. cereus* in dairy products in China and provided the analytical assays of *B. cereus* and its toxins. PCR is the most commonly used analysis method, accounting for 49% of articles on *B. cereus* (Figure 4a). However, the expression of genes does not completely represent the toxicity of the virulence factors. We need more comprehensive and rapid testing methods such as cytotoxicity tests or LC-MS analysis. Thus, the role in cell toxicity of the virulence factors of *B. cereus* is unquestionably important, and understanding the actions of *B. cereus* and its toxins at the cellular level would benefit the prevention of *Bacillus* infections. In addition, the cellular mechanism and the interaction between different virulence factors should be further studied. Recent reports showed that the persistent *B. cereus* interaction with host cells is even hard to control [1,49,143], and compared to other persistent bacteria, *B. cereus* are more dangerous due to the high transmission and viability of their spores [69,99,144]. In sum, strong emphasis should be placed on the *B. cereus* in dairy products to guarantee the safety of human life in China.

## Figures and Tables

**Figure 1 toxins-12-00454-f001:**
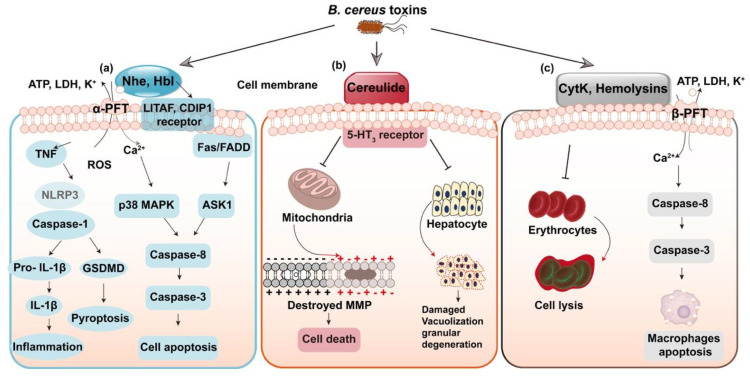
Modes of action of the toxins in *B. cereus*. (**a**) Non-hemolytic enterotoxin (Nhe) and hemolysin BL (Hbl) perforate the cell membrane. Nhe promotes the NLRP3 inflammasome and induces caspase-8 dependent apoptosis. (**b**) Cereulide induces destroyed mitochondrial membrane potential (MMP) and leads to hepatocyte damage. (**c**) CytK and Hemolysins otherwise do harm to the target membrane, leading to cell lysis and cell apoptosis in macrophages.

**Figure 2 toxins-12-00454-f002:**
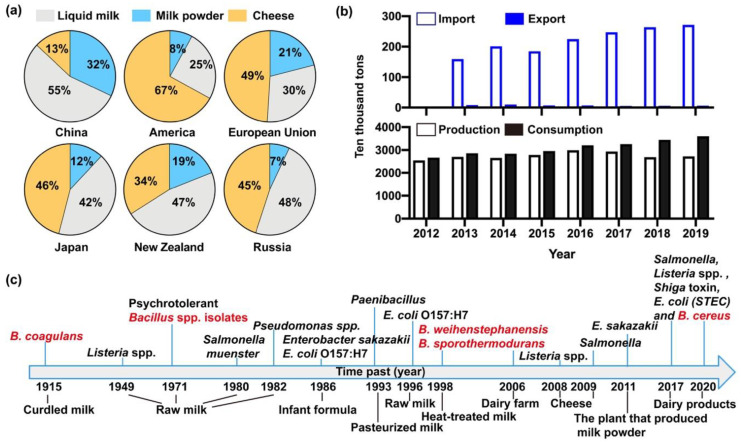
Development of the dairy industry in China and the world. (**a**) The production, consumption, import and export of milk and milk products in China from 2012–2019. Data from National Bureau of Statistics of China. (**b**) The consumption of liquid milk, milk powder and cheese in China, America, European Union, Japan, New Zealand and Russia. (**c**) Microbial outbreaks in raw milk and dairy products during the past century all over the world [64]. Red marked isolates were *B. cereus* spp. strains.

**Figure 3 toxins-12-00454-f003:**
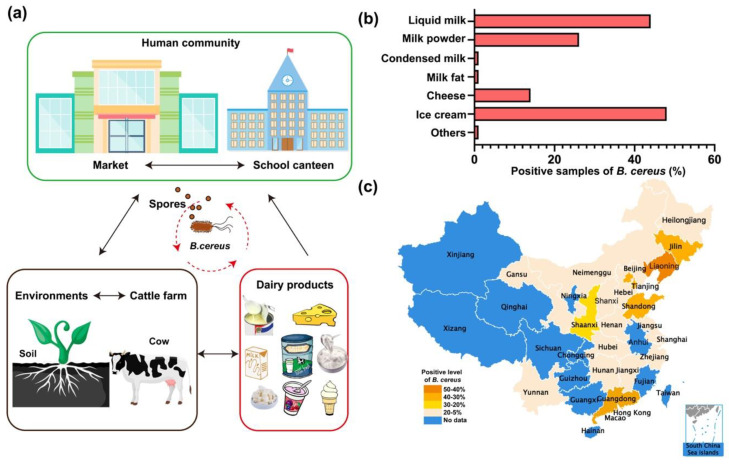
The risk assessment of *Bacillus cereus* in milk and milk products. (**a**) *B. cereus* and spores circularly contaminate human markets, dairy farm and dairy products. (**b**) The red column diagram (lower left corner) represents the positive rate of *Bacillus cereus* in seven groups of milk and milk products. (**c**) The China map shows the regional specificity of *B. cereus* contamination. Blue indicates no reliable data are found in these provinces. There are four different levels of contamination rate and the darker the color is, the rate of *B. cereus* is higher. In particular, the reddish, orange, yellow and pale pink color signify the rate of 50–40%, 40–30%, 30–20% and 20–5%, respectively.

**Figure 4 toxins-12-00454-f004:**
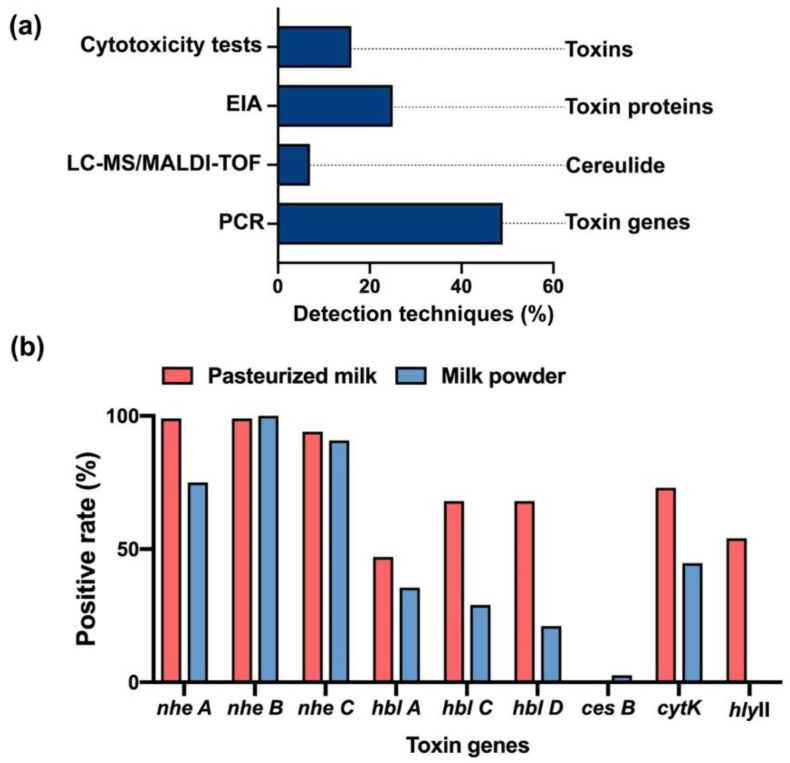
Detection techniques of *B. cereus* and its virulence factors. (**a**) The percentages of articles using each detection technique for *B. cereus* and its virulence factors. These techniques include cytotoxicity tests, EIA, LC-MS/MALDI-TOF and PCR. (**b**) The positive rate of toxin genes in *Bacillus* samples from pasteurized milk or milk powder in China.

**Table 1 toxins-12-00454-t001:** The prevalence of *Bacillus cereus* and its virulence factors from dairy products in China.

Source	Region	Year	No. of *B. cereus* Isolates/ No. of Samples	Detection of Toxin Genes (%)									Reference
				*nheA*	*nheB*	*nheC*	*hblA*	*hblC*	*hblD*	*cesB*	*cytK*	*HlyⅡ*	
Raw milk	Beijing	2013–2014	92/306	100	100	100	79	79	79	ND	ND	ND	[13]
Raw milk	Northeast China	2017–2018	56/350	ND	ND	ND	ND	ND	ND	ND	ND	ND	[77]
Pasteurized milk	Major cities in China, including Beijing, Nanchang, Chengdu, Hefei, Wuhan, Shanghai, et al.	2011–2016	70/258	99	99	94	47	68	68	5	73	54	[14]
Pasteurized milk	Wuhan	2006	26/54	71.7	62	71.7	37	66.3	71.7	ND	ND	ND	[80]
Ice cream	Wuhan	2006	24/40	ND	ND	ND	ND	ND	ND	ND	ND	ND	[66]
Milk powder	Wenzhou	2015–2016	76/400	75	100	90.8	35.5	29.0	21.1	ND	44.7	ND	[81]
Infant formula	Chinese markets	2012–2013	74/513	ND	ND	ND	ND	ND	ND	ND	ND	ND	[82]
Infant formula	Chinese markets	2015	57/135	87.7	87.7	49.1	24.6	22.8	17.5	3.5	22.8	ND	[83]
Infant formula	Liaoning	2016	22/176	90.9	72.7	100	0	59.1	54.5	ND	68.2	ND	[84]
Infant formula	Chinese markets	2013–2015	33/401	ND	ND	ND	ND	ND	ND	ND	ND	ND	[7]
Infant formula	Liaoning	2016–2017	70/166	ND	ND	ND	ND	ND	ND	ND	ND	ND	[76]
Infant formula	Yunnan	2012–2016	71/605	ND	ND	ND	ND	ND	ND	ND	ND	ND	[85]
Infant formula	Kunming	2016	5/126	ND	ND	ND	ND	ND	ND	ND	ND	ND	[86]
Infant formula and processing facility	Gansu	2013–2014	31/183	ND	ND	ND	ND	ND	ND	ND	ND	ND	[87]
Infant formula	Heilongjiang, Hebei, Henan, Hubei, Hunan, Jiangsu, Jiangxi, Guangdong	2012	115/817	ND	ND	ND	ND	ND	ND	ND	ND	ND	[88]
Dairy products	Heilongjiang, Jilin, Hebei, Henan, Guizhou	2018–2019	54/500	94.4	94.4	100	57.4	68.5	16.7	11.1	75.9	53.7	[12]

**Table 2 toxins-12-00454-t002:** The distribution of *Bacillus* toxins in dairy products in China and other countries.

Toxin Genes (%)									Origin	Source	Year	Reference
*nheA*	*nheB*	*nheC*	*hblA*	*hblC*	*hblD*	*cesB*	*cytK*	*hlyⅡ*				
*nhe* 100			*hbl* 78.3			1.1	--	--	Beijing, China	Dairy farms	2013–2014	[13]
90.9	72.7	100	0	59.1	54.5	--	68.2	--	Liaoning, China	Milk powder	2016	[84]
87.2	81.6	86.4	36	38.4	38.4	3.2	36.8	--	Hebei, Hainan, Yunnan province, et al., China	Milk powder	2019	[129]
74.1	88.9	100	55.6	77.8	0	48.2	33.3	--	China	UHT milk processing line	2014–2015	[130]
84.1	89.9	84.1	59.4	44.9	53.6	2.9	53.6	--	Brazil	Dairy production chain	2016	[62]
*nhe* 100			*hbl* 29.5			0	24.1	--	Colombia	Ready-to-eat food and milk	2013	[131]
76.5	--	--	--	41.2	--	0	5.9	--	Canada	Pasteurized milk	2014–2015	[132]
96	99	100	44	40	44	--	42	23	France	FBO	2007–2014	[133]
6.3	2.1	4.2	11.5	10.4	16.7	9.4	75	--	Ghana	Dairy farm	2015	[73]
60	60	60	13	13	113	--	75	--	Turkey	Milk and cheese	2013	[134]

Note: FBO indicates foodborne outbreak; “—” represents no analytical tests was performed to identify specific toxin genes.

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
