# Peer review of "Characterization of Bacillus cereus in Dairy Products in China"

_toxins, 2020, doi:10.3390/toxins12070454_

Round 1

Reviewer 1 Report

This review aims to provide an overview of the prevalence, detection (methods?), and the toxicity of the B. cereus isolates found in dairy products in China.

Overall, there is a need for a thorough review of this topic; however, this manuscript needs to be improved to be suitable for publication.

Specifically, authors are encouraged to revise the manuscript to fix grammar. Here are a few examples from the introduction: L18 – a verb missing after “that”; L28: encoding should be changed to encoded; L33: take it together should be changed to taken together; L37: analytical assays of B. cereus should be changed to “for detection of B. cereus”). Furthermore, all data reported in a review paper need to be cited, and accurate and comprehensive information on B. cereus group species and their toxin detection need to be reported. Please see the specific comments below.

L44: Increasing needs of human needs?

L46: And so on? Please list all relevant categories if listing categories is relevant to the reviewed topic.

L48: It is not clear what you meant by specific categories having top priority.

L58: Is detection of any microbes important or did you have in mind those that are commonly linked with quality and safety issues?

L60: Pollution may be better phrased as contamination in the context of food.

L61: Could you provide references that support the statement that B. cereus spp. contamination is constantly rising all over the world?

L63: Could you explain why the prevalence of B. cereus spp. is difficult to estimate? What is the evidence that B. cereus spp. outbreaks are a “thorny” problem globally?

L64: Would it be better to say suspected rather than accused?

L65: Please correct the grammar of the sentence. Furthermore, is epidemic an appropriate word for what you are covering in this review? Did you mean epidemiology?

Figure 1: The species marked in red are not just B. cereus group species; they are other Bacillus spp. that are not causing human illness. Some of these species cause milk spoilage, which is not a food safety concern.

L73: Are you reviewing specifically B. cereus sensu strictu incidence or the incidence of any B. cereus group species?

L74: Is an epidemic a correct expression for what you are describing?

L77: B. cereus or B. cereus group species (other than B. cereus s.s.)?

L79: Did you mean cause (rather than incentive)?

L87: Could you provide more references?

Figure 2: Is this figure really showing risk assessment? If so, could you describe the risk more clearly in the figure caption? Does B show a rate or occurrence of B. cereus spp. positive samples?

L96: Could you explain how the volume of milk consumption affects the prevalence of B. cereus spp. in food? I can see how high consumption of milk could possibly result in a higher occurrence of B. cereus spp.-linked foodborne illness; however, it is not clear to me how that would result in a higher occurrence of contamination of milk.

L103: Could you cite the source of these data?

L127: What do you mean by concluding all virulence factors?

L129: Could you explain how the detection of virulence factors could be helpful in the prevention of foodborne illness?

L131: PlcR is not just a Phospholipase C regulator. It is also not responsible for the formation of the listed products, but it is involved in the regulation of the transcription for respective genes.

L137: pOX1 is a plasmid, not a regulator of ces gene cluster.

L143: Why just spiral-plating? Any plating technique, especially spread-plating can be used.

L148: One of the most commonly used loci for molecular detection and subtyping of the B. cereus spp. is not mentioned – the panC gene that is used for the phylogenetic group assignment of B. cereus group isolates. Please review the extensive body of literature in this area and make sure that the synthesis provided in this review reflects the commonly used methods.

L151: PCR-based methods do not allow for the detection of secreted toxins, they detect the genes encoding these toxins. Please revise to ensure accuracy.

L178: Please provide a comprehensive synthesis of the existing literature on the cytotoxicity assessment.

Please cite the source of the data presented in Figure 3 and L179-186.

L203: Could you please explain how Nhe is a non-redundant toxin?

L208: The published research also shows the potential synergistic activity of Nhe with sphingomyelinase enzyme, which is not mentioned here.

Reviewer 2 Report

In the presented manuscript the authors have reviewed the problem of Bacillus contamination in diary products in China. Considering that Bacillus cereus is the main contaminant of diary product and that the contamination might lead to severe diseases, the provided data is very interesting and significant. Nevertheless, the structure of the review should be improved.

Major comments:

Though first three sections provide relevant and interesting information, the section 4 and 5 seems to be strange. The section 4 is a short summary of Bacillus biology and methods of its detection. The only information regarding diary products in China is the short summation of the Table 1, provided in the previous section. The section 5 is just the short review of molecular mechanisms of Bacillus virulence factors. It would be better to provide a distribution of different strains and their virulence factors across China, what is the particulars of strains in diary products all across the world and in China and so on.

Minor comments^

Also, I would recommend to proofread the text more thoroughly. For example, in the text milk powder is called milk power. On line 123 Bacillus cereus is written without capital letter and italization.

Round 2

Reviewer 1 Report

Thank you for addressing the comments.

Author Response

Thank you for your comment.

Reviewer 2 Report

I would like to thank the authors for addressing my comments. Though there are still some comments regarding the structure. 

Comment 1. I agree, that sections 4 and 5 contain important information regarding Bacillus biology. But being close to the end of the review, its connection with the problems of food contamination in China is unclear. So, I suggest incorporating the information from these sections in the introduction. Also, the description of distribution of the virulence factors in China provided on L. 102-105, is in section 3, so it is before the explanation what are these genes.

Comment 2. Considering such detailed information of the virulence factors of Bacillus, it would be interesting to read about distribution of these factors across China in connection with geography, transport particulars and so on. Also, a comparison with situation in other countries would be beneficial.

Comment 3. The description provided on L.193-201 regarding the occurrence rate of the virulence factors, is very brief. It would be better to provide information for other products, provide some information about reasons of such distribution and so on.

Comment 4. The figures are usually placed after the paragraph, where they are mentioned for the first time.

Round 3

Reviewer 2 Report

I would like to thank authors for addressing my comments.